# Towards a learning theory of representation alignment

**Francesco Insulla**
Institute of Computational and Mathematical Engineering
Stanford University
Stanford, CA 94305, USA
`franinsu@stanford.edu`

**Shuo Huang**
Istituto Italiano di Tecnologia
Genoa, GE 16163, Italy
`shuo.huang@iit.it`

**Lorenzo Rosasco**
MaLGa Center, DIBRIS, Università di Genova, Genoa, GE 16146, Italy
CBMM, Massachusetts Institute of Technology, Cambridge, MA 02139, USA
Istituto Italiano di Tecnologia, Genoa, GE 16163, Italy
`lrosasco@mit.edu`

## Abstract

It has recently been argued that AI models' representations are becoming aligned as their scale and performance increase. Empirical analyses have been designed to support this idea and conjecture the possible alignment of different representations toward a shared statistical model of reality. In this paper, we propose a learning-theoretic perspective to representation alignment. First, we review and connect different notions of alignment based on metric, probabilistic, and spectral ideas. Then, we focus on stitching, a particular approach to understanding the interplay between different representations in the context of a task. Our main contribution here is to relate the properties of stitching to the kernel alignment of the underlying representation. Our results can be seen as a first step toward casting representation alignment as a learning-theoretic problem.

## 1 Introduction

In recent years, as AI systems have grown in scale and performance, attention has moved towards universal models that share architecture across modalities. Examples of such systems include CLIP (Radford et al., 2021), VinVL (Zhang et al., 2021), FLAVA (Singh et al., 2022), OpenAI's GPT-4 (OpenAI, 2023), and Google's Gemini (Google, 2023). These models are trained on diverse datasets containing both images and text and yield embeddings that can be used for downstream tasks in either modality or for tasks that require both modalities. The emergence of this new class of multi-modal models poses interesting questions regarding alignment and the trade-offs between unimodal and multimodal modeling. While multimodal models may provide access to greater scale through dataset size and computational efficiency, how well do features learned from different modalities correspond to each other? How do we mathematically quantify and evaluate this alignment and feature learning across modalities?

Regarding alignment, Huh et al. (2024) observed that as the scale and performance of deep networks increase, the models' representations tend to align. They further conjectured that the limiting representations accurately describe reality - known as *Platonic representation hypothesis*. Their analysis also suggests that alignment correlates with performance, implying that improving the alignment of learned features across different modalities could enhance a model's generalization ability. However, alignment across modalities has yet to be evaluated in a more interpretable manner, and theoretical guarantees of alignment under realistic assumptions are still lacking.

One way to quantify alignment is by kernel alignment, introduced by Cristianini et al. (2001), which evaluates the correlation of two kernel matrices $K_{1,n}, K_{2,n}$ through Frobenius norms

$$\widehat{A}(K_{1,n}, K_{2,n}) = \frac{\langle K_{1,n}, K_{2,n} \rangle_F}{\sqrt{\langle K_{1,n}, K_{1,n} \rangle_F \langle K_{2,n}, K_{2,n} \rangle_F}}.$$

Following this direction, methods such as Centered Kernel Alignment (CKA) (Kornblith et al., 2019) and Singular Vector Canonical Correlation Analysis (SVCCA) (Raghu et al., 2017) were developed to compare the learned representations. Another class of metrics is derived from independence testing, including the Hilbert-Schmidt Independence Criterion (HSIC) (Gretton et al., 2005a) and Mutual Information (MI) (Hjelm et al., 2019). However, further research is needed to clarify the relationships among these methods and other frameworks for assessing alignment.

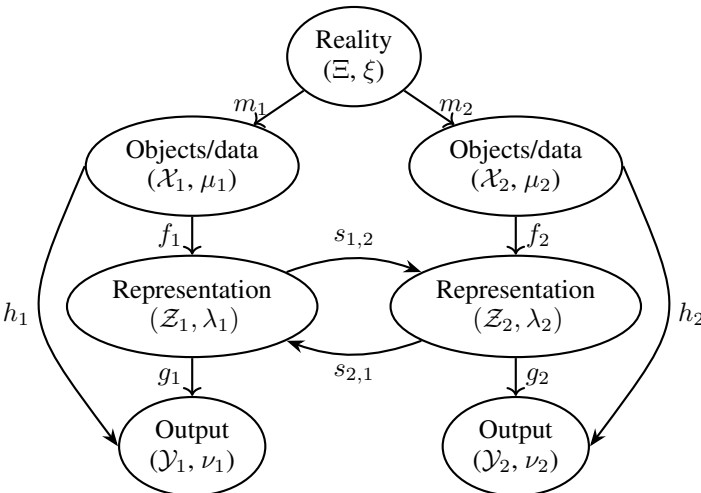

Figure 1: Diagram illustrating the process of multi-modal learning. It contains spaces and measures of reality, objects/data, representation, and outputs as well as the functions connecting them. A detailed explanation of these symbols is in Section 2.

To quantify the alignment of representation conditioned on a task, one approach is to use the stitching method (Lenc & Vedaldi, 2015). Bansal et al. (2021) revisited this technique and used it to highlight that good models trained on different objectives (supervised vs self-supervised) have similar representations. By evaluating how well one representation integrates into another model, stitching provides a more interpretable framework for assessing alignment. To describe the setup, we use $h_{1,2} := g_2 \circ s_{1,2} \circ f_1$ to represent the function after stitching from model 1 to model 2 (Figure 1 gives a detailed illustration of the whole process). Here $g_q$ and $f_q$ are parts of model $\mathcal{H}_q : \mathcal{X}_q \to \mathcal{Y}_q$ with $q = 1, 2$, and $s_{1,2}$ is the stitcher. We consider the generalization error after stitching between two models:

$$\mathcal{R}(g_2 \circ s_{1,2} \circ f_1) = \mathbb{E}\left[\ell(h_{1,2}(x), y)\right].$$

We can use the risk of the stitched model in excess of the risk of model 2

$$\min_{s_{1,2}} \mathcal{R}(h_{1,2}) - \mathcal{R}(h_2)$$

to quantify the impact of using different representations, fixing $g_2$.

In this paper, we aim to formalize and refine some of these questions, and our contributions are summarized as follows:

(a) We compile different definitions of alignment from various communities, demonstrate their connections, and give spectral interpretations.

  • Starting from the empirical Kernel Alignment (KA), we reformulate empirical KA and population version of KA using feature/representation maps, operators in Reproducing Kernel Hilbert Space (RKHS), and spectral interpretation. In addition, we discuss the statistical properties of KA.

- We integrate various notions of alignment, ranging from kernel alignment in independence testing and learning theory to measure and metric alignment, and demonstrate their relationships and correlations. This comprehensive exploration provides a deeper understanding for practical applications.

(b) We provide a generalization error bound of linear stitching with the kernel alignment of the underlying representation.

- A linear $g_q$ results in the stitching error being equivalent to the risk from the model $\mathcal{H}_q$. This occurs, for example, when $\mathcal{H}_q$ represents RKHSs or neural networks, then $g_q$ is a linear combination of features in RKHSs or the output linear layers of neural networks.
- The excess stitching risk can be bounded by kernel alignment when $g_q$ are nonlinear functions with the Lipschitz property. A typical scenario is stitching across the intermediate layers of neural networks.
- For models involving several compositions, such as deep networks, if we stitch from a layer further from the output to a layer closer to the output (stitching forward) and $g_q$ is Lipschitz, the difference in risk can be bounded by stitching.

**Structure of the paper**    In the following of this paper, we introduce the problem settings and some notation in Section 2. Different definitions for representation alignment from different communities and the relationship among them will be derived in Section 3. Section 4 demonstrates that the stitching error could be bounded by the kernel alignment metric. And the conclusion is given in Section 5.

## 2 Preliminaries

Empirical results demonstrate that well-aligned features significantly enhance task performance. However, there is a pressing need for more rigorous mathematical tools to formalize and quantify these concepts in uni/multi-modal settings. In this section, we provide a mathematical formalization of uni/ multi-modal learning, introducing key notation to facilitate a deeper understanding of the underlying processes.

**Setup**    Without loss of generality, we focus on the case of two modalities, as illustrated in Figure 1, which outlines the corresponding process. For $q = 1, 2$, let $(\mathcal{X}_q, \mu_q)$ and $(\mathcal{Z}_q, \lambda_q)$ be probability spaces, and let $\mathcal{F}_q$ be spaces of functions $f_q : \mathcal{X}_q \to \mathcal{Z}_q = \mathbb{R}^{d_q}$. We regard $\mathcal{X}_q$ as the space of **objects** (or data), $\mathcal{F}_q$ as the space of representation (or embedding) maps, and $\mathcal{Z}_q$ as the space of **representations**. We relate $\mu_q$ and $\lambda_q$ by assuming $\lambda_q = (f_q)_{\#}\mu_q$[1]. We also assume that $\mu_1$ and $\mu_2$ are the marginals of a joint probability space $(\mathcal{X}, \mu)$ with $\mathcal{X} = \mathcal{X}_1 \times \mathcal{X}_2, \quad \mu_q = (\pi_q)_{\#}\mu$, where $\pi_q : \mathcal{X} \to \mathcal{X}_q$ is the projection map. Moreover, let $(\mathcal{Y}_q, \nu_q)$ be the task-based **output** spaces and define $\mathcal{G}_q = \{g_q : \mathcal{Z}_q \to \mathcal{Y}_q\}$ with $\nu_q = (g_q)_{\#}\lambda_q$. Each overall model is generated by $\mathcal{H}_q := \{h_q : \mathcal{X}_q \to \mathcal{Y}_q \mid h_q = g_q \circ f_q\}$.

**Reality**    Consider a space of abstract objects, called the *reality space* and denoted by $\Xi$, which generates the observed data in various modalities through maps $m_q : \Xi \to \mathcal{X}_q$. These maps may be bijective, lossy, or stochastic. Reality can be modeled as a probability space $(\Xi, \xi)$. Alternatively, one may define reality as the joint distribution over modalities by setting $m_q = \pi_q$.

**Uni/Multi-modal**    We may want to consider the case of a single modality, where only one data space exists, versus multiple modalities, where several such spaces are present. Two modalities are deemed equal if $\pi_1 = \pi_2$.

**Representation alignment**    A representation mapping is a function $f : \mathcal{X} \to \mathbb{R}^d$ that assigns a latent feature vector in $\mathbb{R}^d$ to each input in the data domain $\mathcal{X}$. Alignment provides a metric to evaluate how well the latent feature spaces obtained from different representation mappings, whether

---

[1] $(f_q)_{\#}\mu_q$ is the pushforward measure of $\mu_q$ defined as $(f_q)_{\#}\mu_q(A) = \mu_q(f_q^{-1}(A))$. In terms of random variables $X_q$ and $Z_q$ with measures $\mu_q$ and $\lambda_q$, respectively, this is equivalent to $f_q(X_q)$ and $Z_q$ being equal in law.

from uni-modal or multi-modal data, are aligned or similar. Commonly used metrics for quantifying alignment include those derived from kernel alignment, contrastive learning, mutual information, canonical correlation analysis, and cross-modal mechanisms, among others. However, they are introduced in a very fragmented manner, without an integrated or unified concept. A detailed introduction and analysis of these methods will be provided in Section 3.

## 3 FRAMEWORKS FOR REPRESENTATION ALIGNMENT

In this section, we describe various definitions of representation alignment from different communities and demonstrate the relationship among them. We begin with a detailed presentation of empirical and population Kernel Alignment and its statistical properties. We then cover other notions of alignment coming from metrics, independence testing, and probability measures, as well as their spectral interpretations. We draw connections to kernel alignment which emerges as a central object.

### 3.1 KERNEL ALIGNMENT (KA)

Based on the work of Cristianini et al. (2001), who introduced the definition of kernel alignment using empirical kernel matrices, we propose different perspectives to understand kernel alignment in both empirical and population settings and derive its statistical properties accordingly.

A reproducing positive definite kernel $K : \mathcal{X} \times \mathcal{X} \to \mathbb{R}$ captures the notion of similarity between objects by inducing an inner product in the associated reproducing kernel Hilbert space (RKHS) $\mathcal{H}$. Specifically, $K(x, x') = \langle f(x), f(x') \rangle$ for any representation (feature) map $f \in \mathcal{H}$, and $x, x' \in \mathcal{X}$. For the multi-modal case, we define $K_q(x, x') := \tilde{K}_q(\pi_q(x), \pi_q(x')) = \tilde{K}_q(x_q, x'_q)$, where $\tilde{K}_q$ is the reproducing kernel associated with $\mathcal{H}_q$. In other words, $K_q$ acts on $x = (x_1, x_2)$ by first applying the projection $\pi_q(x) = x_q$. In the following, the subscript $x_q$ denotes the $q$-th modality, and the superscript $x^i$ indicates the $i$-th sample.

#### 3.1.1 EMPIRICAL KA

From Cristianini et al. (2001), we adopt the following formulation for kernel alignment for kernel matrix $K_{q,n} \in \mathbb{R}^{n \times n}$ with samples $\{x^i\}_{i=1}^n$ drawing according to the probability measure $\mu$

$$\widehat{A}(K_{1,n}, K_{2,n}) = \frac{\langle K_{1,n}, K_{2,n} \rangle_F}{\sqrt{\langle K_{1,n}, K_{1,n} \rangle_F \langle K_{2,n}, K_{2,n} \rangle_F}},$$

where $\langle K_{1,n}, K_{2,n} \rangle_F = \sum_{i,j=1}^n K_{1,n}(x^i, x^j) K_{2,n}(x^i, x^j)$. One modification is to first demean the kernel by applying a matrix $H = I_n - \frac{1}{n} 1_n 1_n^T$ on the left and right of each $K_{q,n}$ with $I \in \mathbb{R}^{n \times n}$ being the identity matrix and $1_n$ being the ones vectors. This results in Centered Kernel Alignment (CKA).

**Representation interpretation of KA**    Denote the empirical cross-covariance matrix between the representation maps $f_1$ and $f_2$ as $\widehat{\Sigma}_{1,2} = \mathbb{E}_n \left[ f_1(x) f_2(x)^T \right] = \frac{1}{n} \sum_{i=1}^n f_1(x^i) f_2(x^i)^T \in \mathbb{R}^{d_1 \times d_2}$. Then the empirical KA will become

$$\widehat{A}(K_{1,n}, K_{2,n}) = \frac{\|\widehat{\Sigma}_{1,2}\|_F^2}{\|\widehat{\Sigma}_{1,1}\|_F \|\widehat{\Sigma}_{2,2}\|_F}. \tag{1}$$

**RKHS operator interpretation of KA**    Inspired by the equation 1, we construct a consistent definition of Kernel Alignment using tools of RKHS, where it suffices to consider output in one dimension[2]. Consider RKHS $\mathcal{H}_q$ containing functions $h_q : \mathcal{X}_q \to \mathbb{R}$ with kernel $K_q$. Given evaluation (sampling) operators $\widehat{S}_q : \mathcal{H}_q \to \mathbb{R}^n$ defined by $(\widehat{S}_q h_q)^i = h_q(x_q^i) = \langle h_q, K_{q,x_q^i} \rangle$. It is not hard to check that the adjoint operator $\widehat{S}_q^* : \mathbb{R}^n \to \mathcal{H}_q$ can be written as $\widehat{S}_q^*(w^1, \ldots, w^n) =$

---

[2]We can generalize the definition to vector-valued functions by recasting $h_q : \mathcal{X}_q \to \mathbb{R}^{t_q}$ as $h_q : \mathcal{X}_q \times [t_q] \to \mathbb{R}$ i.e. with kernels of the form $K_q(x_q, i, x'_q, i')$ for integers $1 \le i, i' \le t_q$.

$\sum_{i=1}^{n} w^i K_q(x_q^i, \cdot)$ and the empirical kernels can be written as $K_{q,n}/n = \widehat{S}_q \widehat{S}_q^*$ (De Vito et al., 2005; Smale & Zhou, 2007). Then the empirical KA may be written as

$$\widehat{A}(K_{1,n}, K_{2,n}) = \frac{\langle \widehat{S}_1 \widehat{S}_1^*, \widehat{S}_2 \widehat{S}_2^* \rangle_F}{\|\widehat{S}_1 \widehat{S}_1^*\|_F \|\widehat{S}_2 \widehat{S}_2^*\|_F} = \frac{\|\widehat{S}_1^* \widehat{S}_2\|_F^2}{\|\widehat{S}_1^* \widehat{S}_1\|_F \|\widehat{S}_2^* \widehat{S}_2\|_F},$$

where $\widehat{S}_1^* \widehat{S}_2 = \frac{1}{n} \sum_i K_{1,x_1^i} \otimes K_{2,x_2^i}$ and it coincides with the literature about learning theory with RKHS.

### 3.1.2 POPULATION VERSION OF KA

For the population setting (infinite data limit of the evaluation operator) in $L^2$, the restriction operator $S_q : \mathcal{H}_q \to L^2(\mathcal{X}_q, \mu)$ is defined by $S_q h_q(x) = \langle h_q, K_q(x, \cdot) \rangle_{K_q}$ and its adjoint $S_q^* : L^2(\mathcal{X}_q, \mu) \to \mathcal{H}_q$ is given by $S_q^* g = \int_{\mathcal{X}} g(x) K_q(x, \cdot) dx$. Then the integral operator $L_{K_q} = S_q S_q^* : L^2(\mathcal{X}_q, \mu) \to L^2(\mathcal{X}_q, \mu)$ is given by $L_{K_q} g(x) = \int_{\mathcal{X}} K_q(x, x') g(x') d\mu(x')$ and the operator $\Sigma_q = S_q^* S_q : \mathcal{H}_q \to \mathcal{H}_q$ can be written as $\Sigma_q = \int_{\mathcal{X}} K_q(x, \cdot) \otimes K_q(x, \cdot) d\mu(x)$ (De Vito et al., 2005; Rosasco et al., 2010). Similarly, the population KA between two kernels $K_1, K_2$ can be defined by

$$A(K_1, K_2) = \frac{\mathrm{Tr}(L_{K_1} L_{K_2})}{\sqrt{\mathrm{Tr}(L_{K_1}^2) \mathrm{Tr}(L_{K_2}^2)}},$$

where the summation in $\langle K_{1,n}, K_{2,n} \rangle_F$ becomes the integration as

$$\mathrm{Tr}(L_{K_1} L_{K_2}) = \int d\mu(x_1, x_2) d\mu(x_1', x_2') K_1(x_1, x_1') K_2(x_2, x_2').$$

If $K_q(x, x') = \langle f_q(x), f_q(x') \rangle$, then $S_q^* S_q$ is a projection onto the span of coordinates of $f_q$. The population version of CKA is KA with $S_q$ replaced with $HS_q$.

**Spectral Interpretation of KA** The understanding of kernel alignment (KA) can be deepened via the spectral decomposition of the associated integral operator. The mercer kernel $K$ can be decomposed as $K = \sum_i \eta_i \phi_i \otimes \phi_i$, where $\eta_i$ are the eigenvalues and $\phi_i$ are the eigenfunctions of the integral operator $L_K$ (Cucker & Smale, 2002; Schölkopf, 2002). Defining the features as $f_i = \sqrt{\eta_i} \phi_i$ and expressing the target function as $h = \sum_i w_i f_i$, we obtain

$$A(K, h \otimes h) = \frac{\sum_i \eta_i^2 w_i^2}{\sqrt{\sum_i \eta_i^2 \sum_i \eta_i w_i^2}}.$$

Similarly, given two kernels $K_q = \sum_i \eta_{q,i} \phi_{q,i} \otimes \phi_{q,i}$ with $f_{q,i} = \sqrt{\eta_{q,i}} \phi_{q,i}$, we have

$$A(K_1, K_2) = \frac{\sum_{i,j} \langle f_{1,i}, f_{2,j} \rangle}{\sqrt{\sum_i \eta_{1,i}^2 \sum_i \eta_{2,i}^2}} = \frac{\sum_{i,j} \eta_{1,i} \eta_{2,j} \langle \phi_{1,i}, \phi_{2,j} \rangle^2}{\sqrt{\sum_i \eta_{1,i}^2 \sum_i \eta_{2,i}^2}}.$$

Letting $[C_{1,2}]_{i,j} = \langle \phi_{1,i}, \phi_{2,j} \rangle$ and defining $\hat{\eta}_i = \eta_i / \|\eta_i\|$, we can equivalently write

$$A(K_1, K_2) = \mathrm{Tr}\left[ C_{1,2} \, \mathrm{diag}(\hat{\eta}_2) \, C_{1,2}^T \, \mathrm{diag}(\hat{\eta}_1) \right] = \langle \hat{\eta}_1, (C_{1,2} \odot C_{1,2}) \hat{\eta}_2 \rangle = \langle \hat{\eta}_1 \hat{\eta}_2^T, C_{1,2} \odot C_{1,2} \rangle$$

with $\odot$ as the Hadamard product. This formulation provides insight into kernel alignment by relating it to the similarity between the eigenfunctions of the two integral operators. In particular, if $\eta_1$ and $\eta_2$ are constant, then $A(K_1, K_2) \propto \|C_{1,2}\|^2$; and if $C_{1,2} = I$, then $A(K_1, K_2) = \langle \hat{\eta}_1, \hat{\eta}_2 \rangle$.

### 3.1.3 STATISTICAL PROPERTIES OF KA.

Having introduced both the empirical and population versions of KA, we now explore its statistical properties. Cristianini et al. (2006) shows that empirical KA concentrates to its expectation by McDiarmid's inequality and gives an $O(1/\sqrt{n})$ bound. For completeness, we state the following lemma summarizing this statistical property and the proof is provided in Appendix 6.3.

**Lemma 1.** *Let $K_1, K_2$ be two kernels for different representations and $\widehat{K}_{1,n}, \widehat{K}_{2,n} \in \mathbb{R}^{n \times n}$ be kernel matrices generated by $n$ samples, then with probability at least $1 - \delta$, we have*

$$\widehat{A}(K_{1,n}, K_{2,n}) - A(K_1, K_2) \leq \sqrt{(32/n) \log(2/\delta)}.$$

## 3.2 ALIGNMENT FROM DISTANCE ALIGNMENT

**Distance alignment (DA)** Given distances $d_q : \mathcal{X}_q \times \mathcal{X}_q \to \mathbb{R}$, then we can compare the difference of two spaces by

$$D(d_1, d_2) = \int (d_1^2(x, x') - d_2^2(x, x'))^2 d\mu(x) d\mu(x').$$

**Equivalence between KA and DA** Suppose $d_q^2 = 2(1 - K_q)$ (Igel et al., 2007) and $K_q(x_q, x_q) = 1$, which emerges naturally from assuming $K_q(x, x') = \langle f_q(x), f_q(x') \rangle, \|f_q(x)\| = 1$, and $d_q^2(x_q, x_q') = \|f_q(x_q) - f_q(x_q')\|^2$, (i.e., $K_q$ represents a mapping onto a ball). Also assume $\|K_q\| = C$. Then, $D(d_1, d_2) = 8C(1 - A(K_1, K_2))$, hence the two paradigms are equivalent.

## 3.3 ALIGNMENT FROM INDEPENDENCE TESTING

Independence testing is a statistical method used to assess the degree of dependence between variables. It often involves examining the covariance and correlations between random variables and can also be applied to quantify kernel-based independence. In this section, we outline several approaches from independence testing within the alignment framework and investigate their connections to the kernel alignment method discussed earlier.

**Hilbert-Schmidt Independence Criterion (HSIC)** The cross-covariance operator for two functions (Baker, 1973) is given by $C_{1,2}[h_1, h_2] = \mathbb{E}_{x_1, x_2}[(h_1(x_1) - \mathbb{E}_{x_1}(h_1(x_1))(h_2(x_2) - \mathbb{E}_{x_2}(h_2(x_2))]$ for $h_1 \in \mathcal{H}_1, h_2 \in \mathcal{H}_2$. From Gretton et al. (2005a)

$$\text{HSIC}(\mu, \mathcal{H}_1, \mathcal{H}_2) = \|C_{1,2}\|_{HS}^2,$$

where $\mu$ is the joint distribution of $\mathcal{X}_1$ and $\mathcal{X}_2$. We can also note that

$$\text{HSIC}(\mu, \mathcal{H}_1, \mathcal{H}_2) = \|\mathbb{E}[K_{x_1} \otimes K_{x_2}]\|^2 = \|\Sigma_{1,2}\|_{HS}^2.$$

Hence HSIC is effectively and unnormalized version of CKA, or, more explicitly,

$$\text{CKA}(K_1, K_2) = \frac{\text{HSIC}(\mathcal{H}_1, \mathcal{H}_2)}{\sqrt{\text{HSIC}(\mathcal{H}_1, \mathcal{H}_1)\text{HSIC}(\mathcal{H}_2, \mathcal{H}_2)}}.$$

**Statistical property of HSIC** Gretton et al. (2005b) shows that, excluding the $O(n^{-1})$ diagonal bias, centered empirical HSIC concentrates to population and Song et al. (2012) provides an unbiased estimator of HSIC and shows its concentration, both by U-statistic arguments.

*Remark* 1 (Other notions from independence testing). There are other concepts of independence testing for alignment such us Constrained Covariance (COCO) (Gretton et al., 2005a), Kernel Canonical Correlation (KCC), Kernel Mutual Information (KMI) (Bach & Jordan, 2002). They are also related to kernel alignment and more detailed explanations can be found in Appendix 6.2.

## 3.4 ALIGNMENT FROM MEASURE ALIGNMENT

There are several methods for comparing measures on the same space. One can then quantify independence by comparing a joint measure with the product of its marginals. This principle allows us to interpret HSIC as test for independence given two function classes.

**MMD to HSIC** Following Gretton et al. (2012), we start by introducing the so-called Maximum Mean Discrepancy (MMD). Let $\mathcal{H}$ be a class of functions $h : \mathcal{X} \to \mathbb{R}$ and let $\mu_q$ be different measures on $\mathcal{X}$. Then, letting $x_q \sim \mu_q$,

$$\text{MMD}(\mu_1, \mu_2; \mathcal{H}) = \sup_{h \in \mathcal{H}} \mathbb{E}[h(x_1) - h(x_2)].$$

Let $\mathcal{H}$ be an RKHS and restrict to a ball of radius 1, then

$$\text{MMD}(\mu_1, \mu_2; \mathcal{H})^2 = \|\mathbb{E}[K_{x_1} - K_{x_2}]\|_{\mathcal{H}}^2 = \mathbb{E}[K(x_1, x_1') + K(x_2, x_2') - 2K(x_1, x_2)].$$

Now we construct a measure of independence by applying MMD on $\mu$ versus $\mu_1 \otimes \mu_2$ where $\mathcal{H}$ is replaced with $\mathcal{H}_1 \times \mathcal{H}_2$ and get HSIC

$$\text{MMD}(\mu, \mu_1 \otimes \mu_2; \mathcal{H}_1 \otimes \mathcal{H}_2)^2 = \text{HSIC}(\mu, \mathcal{H}_1, \mathcal{H}_2) = \|\Sigma_{1,2}\|^2 = \sum_i \rho_i^2$$

where $\{\rho_i^2\}$ is the spectrum of $\Sigma_{1,2}\Sigma_{2,1}$.

We can also use tests of independence that don't explicitly depend on a function class, such as mutual information, by letting $\mu$ be a Gaussian Process measure on two functions in their respective RKHS with covariance defined by their kernels.

**KL Divergence to Mutual Information**  Given KL divergence

$$D_{\text{KL}}(\mu||\nu) = \int d\mu(x) \log\left(\frac{d\mu}{d\nu}(x)\right),$$

we can define mutual information as

$$I(\mu) = D_{\text{KL}}(\mu||\mu_1 \otimes \mu_2) = \int d\mu(x_1, x_2) \log\left(\frac{\mu(x_1, x_2)}{\mu_1(x_1)\mu_2(x_2)}\right) = \int d\mu(x_1, x_2) \log\left(\frac{\mu(x_2|x_1)}{\mu_2(x_2)}\right).$$

For multivariate Gaussian $\mu$, with marginals $\mu_q = \mathcal{N}(0, \Sigma_q)$,

$$\text{MI}(\nu) = \frac{1}{2}\log\left(\frac{|\Sigma_1||\Sigma_2|}{|\Sigma|}\right) = \frac{1}{2}\log\left(\frac{|\Sigma_2|}{|\Sigma_2 - \Sigma_{2,1}\Sigma_1^{-1}\Sigma_{1,2}|}\right).$$

For the simplest case of $\Sigma_q = I$, then this simplifies to

$$\text{MI}(\nu) = -\tfrac{1}{2}\log(|I - \Sigma_{1,2}\Sigma_{2,1}|) = -\tfrac{1}{2}\sum_i \log(1 - \rho_i^2).$$

**Wasserstein distance**  For the Wasserstein distance

$$W_2(\mu, \nu) = \inf\{\mathbb{E}_{(x,y)\sim\gamma}\left[\|x - y\|^2\right] : \gamma_1 = \mu, \gamma_2 = \nu\},$$

applying $\mu$ and $\mu_1 \otimes \mu_2$ to measure independence, we have

$$W_2(\mu, \mu_1 \otimes \mu_2) = \inf\{\mathbb{E}_{((x_1,x_2),(x_1',x_2'))\sim\gamma}\left[\|x_1 - x_1'\|^2 + \|x_2 - x_2'\|^2\right] : \gamma_1 = \mu, \gamma_2 = \mu_1 \otimes \mu_2\}.$$

For mean zero Gaussians

$$W_2(\mu_1, \mu_2) = \text{Tr}[\Sigma_1 + \Sigma_2 - 2(\Sigma_1^{1/2}\Sigma_2\Sigma_1^{1/2})^{1/2}]$$

and as a measure of independence with $\Sigma_q = I$

$$W_2(\mu, \mu_1 \otimes \mu_2) = 2\text{Tr}[I - (I - \Sigma_{1,2}\Sigma_{2,1})^{1/2}] = 2\sum_i \left(1 - \sqrt{1 - \rho_i^2}\right).$$

In summary, we've introduced several popular metrics for alignment between two representations and related them via spectral decompositions to a central notion of kernel alignment generalized for RKHS. Moreover, similar notions can be used to quantify alignment between a model and a task to estimate generalization error, and more details are provided in the Appendix 6.1.

## 4  STITCHING: TASK AWARE REPRESENTATION ALIGNMENT

Building on our understanding of kernel alignment—a fundamental metric for evaluating the alignment of representations detailed in the previous section—we now explore stitching, a task-aware concept of alignment. Stitching involves combining layers or components from various models to create a new model which can be used to understand of how different parts contribute to overall performance or to compare the learned features for a task. In this section, we mathematically formulate this process and provide some intuition by demonstrating that the generalization error after stitching can be bounded by kernel alignment using spectral arguments.

### 4.1 STITCHING ERROR BETWEEN MODELS

In the following, we focus on stitching between two modalities. Figure 1 provides a detailed illustration of the functions, spaces, and compositions in question. Denote the function space for task learning as $\mathcal{H}_q := \{h_q : \mathcal{X}_q \to \mathcal{Y}_q | h_q = g_q \circ f_q, \ g_q \in \mathcal{G}_q, f_q \in \mathcal{F}_q\}$ with $q = 1, 2$. Here $\mathcal{F}_q : \mathcal{X}_q \to \mathcal{Z}_q$ and $\mathcal{G}_q : \mathcal{Z}_q \to \mathcal{Y}_q$. Denote $\mathcal{S}_{1,2} := \{s_{1,2} : \mathcal{Z}_1 \to \mathcal{Z}_2\}$ as the stitching map from $\mathcal{Z}_1$ to $\mathcal{Z}_2$ and $\mathcal{S}_{2,1} := \{s_{2,1} : \mathcal{Z}_2 \to \mathcal{Z}_1\}$ reversely. Define the risk concerning the least squares loss as

$$\mathcal{R}_q(h_q) = \mathbb{E}\left[\|h_q(x) - y\|^2\right] = \int_{\mathcal{X}_q \times \mathcal{Y}_q} \|h_q(x) - y\|^2 \mathrm{d}\rho_q(x, y), \quad h_q \in \mathcal{H}_q.$$

Here, $\rho_q(x, y)$ is the joint distribution of $\mathcal{X}_q$ and $\mathcal{Y}_q$ and we use the notation $\| \cdot \|$ to represent $\| \cdot \|_{\mathcal{Y}_q}$ associated with space $\mathcal{Y}_q$ for simplicity, i.e. absolute value for $\mathcal{Y}_q = \mathbb{R}$, $l_2$ norm for $\mathcal{Y}_q = \mathbb{R}^{t_q}$ and $L_2$ norm for $\mathcal{Y}_q$ being the function space. For $h_q \in \mathcal{H}_q$, denote any minimizer of $\mathcal{R}(h_q)$ among $\mathcal{H}_q$ as $h_q^*$, that is,

$$\mathcal{R}_q(\mathcal{H}_q) := \mathcal{R}_q(h_q^*) = \min_{h \in \mathcal{H}_q} \mathcal{R}_q(h), \quad q = 1, 2.$$

Moreover, denote the function spaces generated after stitching from $\mathcal{Z}_1$ to $\mathcal{Z}_2$ as

$$\mathcal{H}_{1,2} = \{h_{1,2} = g_2 \circ s_{1,2} \circ f_1 : s_{1,2} \in \mathcal{S}_{1,2}\}$$

and conversely as $\mathcal{H}_{2,1}$.

Lenc & Vedaldi (2015) proposed to describe the similarity between two representations by quantifying how usable a representation $f_1$ is when stitching with $g_2$ through a function $s_{1,2} : \mathcal{Z}_1 \to \mathcal{Z}_2$ or oppositely through $s_{2,1} \in \mathcal{S}_{2,1}$. To quantify the similarity, we provide a detailed definition of the stitching error.

**Stitching error**  Define the stitching error as

$$\mathcal{R}_{1,2}^{\mathrm{stitch}}(s_{1,2}) := \mathcal{R}_2(g_2 \circ s_{1,2} \circ f_1) = \mathcal{R}_2(h_{1,2})$$

and the minimum as

$$\mathcal{R}_{1,2}^{\mathrm{stitch}}(\mathcal{S}_{1,2}) := \min_{s_{1,2} \in \mathcal{S}_{1,2}} \mathcal{R}_2(h_{1,2}) = \mathcal{R}_2(\mathcal{H}_{1,2}).$$

To quantify the difference in the use of stitching, we define the **excess stitching risk** as

$$\mathcal{R}_{1,2}^{\mathrm{stitch}}(\mathcal{S}_{1,2}) - \mathcal{R}_2(h_2).$$

Note that $\mathcal{R}_{1,2}^{\mathrm{stitch}}(\mathcal{S}_{1,2}) - \mathcal{R}_2(h_2)$ quantifies a difference in use of representation (fix $g_2$, compare $s_{1,2} \circ f_1$ vs $f_2$), while if $\mathcal{Y}_1 = \mathcal{Y}_2$ and $\mathcal{R}_1 = \mathcal{R}_2$ then $\mathcal{R}_{1,2}^{\mathrm{stitch}}(\mathcal{S}_{1,2}) - \mathcal{R}_1(h_1)$ quantifies difference between $g_2 \circ s_{1,2}$ and $g_1$ (fix $f_1$).

The functions in $\mathcal{S}_{1,2}$ are typically simple maps such as linear layers or convolutions of size one, to avoid introducing any learning, as emphasized in Bansal et al. (2021). The aim is to measure the compatibility of two given representations without fitting a representation to another. One perspective inspired by Lenc & Vedaldi (2015) is that we should not penalize certain symmetries, such as rotations, scaling, or translations, which do not alter the information content of the representations. Furthermore, the amount of unwanted learning may be quantified by stitching from a randomly initialized network.

### 4.2 STITCHING ERROR BOUNDS WITH KERNEL ALIGNMENT

In this section, we focus on a simplified setting where $s_{1,2} : \mathcal{Z}_1 \to \mathcal{Z}_2$ is a **linear stitching**, that is, $s_{1,2}(z_1) = S_{1,2}z_1$ with $S_{1,2} \in \mathbb{R}^{d_2 \times d_1}, z_q \in \mathbb{R}^{d_q}$. Additionally, we assume $\mathcal{Y}_1 = \mathbb{R}^{t_1}, \mathcal{Y}_2 = \mathbb{R}^{t_2}$. In this section, we quantify the stitching error and excess stitching risk using kernel alignment and provide a lower bound for the stitching error when stitching forward.

The following lemma shows that when $\mathcal{G}_q$ are linear, stitching error only measures the difference in risk of $\mathcal{H}_1$ versus $\mathcal{H}_2$.

**Lemma 2.** *Suppose* $dim(\mathcal{Y}_1) = dim(\mathcal{Y}_2) = d$ *and* $\mathcal{R}_1 = \mathcal{R}_2$. *Let* $g_q \in \mathcal{G}_q$ *be linear with* $g_q(z_q) = W_q z_q$ *and* $W_q \in \mathbb{R}^{d \times d_q}$. *Let* $s_{1,2} : \mathcal{Z}_1 \to \mathcal{Z}_2$ *be linear with* $s_{1,2}(z_1) = S_{1,2} z_1$ *and* $S_{1,2} \in \mathbb{R}^{d_2 \times d_1}$. *Then* $\mathcal{R}_{1,2}^{\text{stitch}}(\mathcal{S}_{1,2}) = \mathcal{R}_1(h_1)$.

*Remark* 2. The lemma applies when $\mathcal{H}_q$ represents a neural network with $\mathcal{G}_q$ as the output linear layer, as well as when $\mathcal{H}_q$ is an RKHS with a Mercer kernel and $\mathcal{G}_q$ is the linear map of representations [3].

The next theorem shows the case when $\mathcal{G}_q$ are nonlinear with the $\kappa$-Lipschitz property, $\|g(z) - g(z')\| \leq \kappa \|z - z'\|$. One intermediate example is the stitching between the middle layers of neural networks.

**Theorem 1.** *Suppose* $g_2$ *is* $\kappa_2$-*Lipschitz. Again let* $s_{1,2}$ *be linear, identified with matrix* $S_{1,2}$. *With the spectral interpretations of* $\Sigma_{1,2} = \mathbb{E}\left[f_1 f_2^T\right] = diag(\eta_1)^{1/2} C_{1,2} diag(\eta_2)^{1/2}$ *and* $\tilde{A}_2 = \|I\|_{\eta_2} - \|C_{1,2}\|_{\eta_2}^2$ *as Paragraph 3.1.2, we have*

$$\mathcal{R}_{1,2}^{\text{stitch}}(\mathcal{S}_{1,2}) \leq \mathcal{R}_2(h_2) + \kappa_2^2 \tilde{A}_2 + 2\kappa_2 (\tilde{A}_2 \mathcal{R}_2(h_2))^{1/2}. \tag{2}$$

*Proof.* Breaking $\mathcal{R}_{1,2}^{\text{stitch}}(s_{1,2})$ into two parts and using Cauchy-Schwarz we get

$$\mathbb{E}\left[\|g_2(S_{1,2}f_1)(x) - y\|^2\right]$$
$$=\mathbb{E}\left[\|(g_2(S_{1,2}f_1)(x) - g_2(f_2)(x)) - (y - g_2(f_2)(x))\|^2\right]$$
$$\leq \mathcal{R}_2(h_2) + \mathbb{E}\left[\|g_2(S_{1,2}f_1)(x) - g_2(f_2)(x)\|^2\right] + 2(\mathbb{E}\left[\|g_2(S_{1,2}f_1)(x) - g_2(f_2)(x)\|^2\right]\mathcal{R}_2(h_2))^{1/2}.$$

As $g_2$ is $\kappa_2$-Lipschitz, we can bound with the error from linearly regressing $f_2$ on $f_1$

$$\mathbb{E}\left[\|g_2(S_{1,2}f_1)(x) - g_2(f_2)(x)\|^2\right] \leq \kappa_2^2 \mathbb{E}\left[\|S_{1,2}f_1(x) - f_2(x)\|^2\right]$$
$$= \kappa_2^2(\|S_{1,2}\|_{\eta_1}^2 + \|I\|_{\eta_2}^2 - 2\langle S_{1,2}, \Sigma_{1,2}^T\rangle)$$

with $\|M\|_\eta^2 = \langle M, M\text{diag}(\eta)\rangle$. Taking derivatives, we note that the minimizer of the RHS is $S_{1,2} = \Sigma_{1,2}^T \text{diag}(\eta_1)^{-1}$. Plugging in, the RHS reduces to $\kappa_2^2 \tilde{A}_2$. Thus

$$\mathcal{R}_{1,2}^{\text{stitch}}(\mathcal{S}_{1,2}) \leq \mathcal{R}_{1,2}^{\text{stitch}}(\Sigma_{1,2})$$
$$\leq \mathcal{R}_2(h_2) + \kappa_2^2 \tilde{A}_2 + 2\kappa_2 (\tilde{A}_2 \mathcal{R}_2(h_2))^{1/2}.$$

$\square$

*Remark* 3. In arguing that kernel alignment bounds stitching error for Theorem 1, we made several simplifying assumptions, which we now assess. Firstly, we restricted the stitching $\mathcal{S}_{1,2}$ to linear maps, following the transformations commonly used in practice (Bansal et al., 2021; Csiszárik et al., 2021), and to preserve the significance of the original representations. If we relax this assumption, we observe that a similar result holds, with $\tilde{A}_2 = \inf_{s_{1,2} \in \mathcal{S}_{1,2}} \mathbb{E}[\|s_{1,2}(f_1(x)) - f_2(x)\|^2]$. Interestingly, for $s_{1,2}$ to use only information about the covariance of $f_1, f_2$, similarly to kernel alignment, $s_{1,2}$ must be linear. Furthermore, we note that for stitching classes that include all linear maps, the linear result remains valid.

*Remark* 4. Note that the notion of alignment that appears here, namely $\|I\|_{\eta_2}^2 - \tilde{A}_2 = \|C_{1,2}\|_{\eta_2}^2 = \|C_{1,2}\text{diag}(\eta_2)\|^2$, is similar to, yet distinct from, kernel alignment given by $\|\Sigma_{1,2}\|^2 = \|\text{diag}(\eta_1)^{1/2} C_{1,2} \text{diag}(\eta_2)^{1/2}\|^2$. In particular, the spectrum $\eta_1$ is irrelevant for the bound. However, this does not hold if regularization is added to $S_{1,2}$ by analogy to linear regression.

*Remark* 5. If two representations are similar in the alignment sense, they are also similar in the stitching sense; however, the converse does not necessarily hold. By loose analogy to topology, this suggests that kernel alignment is a stronger notion of similarity.

---

[3] More explicitly, if the RKHS kernel $K_q$ is a sum of separable kernels, then by Mercer's theorem we can decompose it as $K_q = \sum_{\rho=1}^{d_q} \eta_{q,\rho} \phi_{q,\rho} \otimes \phi_{q,\rho}$ where $\eta_{q,\rho} \geq 0$ are the eigenvalues, and $\phi_{q,\rho} : \mathbb{R}^{D_q} \to \mathbb{R}^{d_q}$ are the orthonormal eigenfunctions of the integral operator associated with the kernel $K_q$. Then any $h_q \in \mathcal{H}_q$ can be decomposed as $h_q = g_q \circ f_q$, where $f_q \in \mathcal{F}_q$ is the feature map $f_q(\mathcal{X}_q)_\rho = \sqrt{\eta_\rho} \phi_{q,\rho}(\mathcal{X}_q)$ and $g_q \in \mathcal{G}_q$ is linear $g_q(z_q) = w_q \cdot z_q$.

Excess stitching risk can also serve as an intermediate result to bound the difference in risk. Let $\mathcal{Y}_1 = \mathcal{Y}_2$ and $\mathcal{R}_1 = \mathcal{R}_2$. To obtain a lower bound for $\mathcal{R}_{1,2}^{\text{stitch}}(\mathcal{S}_{1,2})$ in a practical setting, we can assume that $\mathcal{S}_{1,2} \circ \mathcal{G}_2 \subseteq \mathcal{G}_1$. For models involving several compositions, such as deep networks, this condition can hold when stitching from a layer further from the output to a layer closer to the output (i.e., stitching forward), provided that the networks are similar and the layer indices are aligned at the end.

**Lemma 3.** *Let $\mathcal{Y}_1 = \mathcal{Y}_2 = \mathcal{Y}$ and $\mathcal{R}_1 = \mathcal{R}_2 = \mathcal{R}$. If $\mathcal{S}_{1,2} \circ \mathcal{G}_2 \subseteq \mathcal{G}_1$ then $\mathcal{R}_{1,2}^{\text{stitch}}(\mathcal{S}_{1,2}) \geq \mathcal{R}_1(\mathcal{H}_1)$.*

The following theorem derives directly from equation 2 and Lemma 3.

**Theorem 2.** *Let $\mathcal{Y}_1 = \mathcal{Y}_2$ and $\mathcal{R}_1 = \mathcal{R}_2 = \mathcal{R}$. Assume $\mathcal{S}_{1,2} \circ \mathcal{G}_2 \subseteq \mathcal{G}_1$, $g_2$ is $\kappa_2$-Lipschitz, and $\mathcal{R}(h_2) = \mathcal{R}(\mathcal{H}_2)$. Then*

$$\mathcal{R}(\mathcal{H}_1) - \mathcal{R}(\mathcal{H}_2) \leq \mathcal{R}_{1,2}^{\text{stitch}}(\mathcal{S}_{1,2}) - \mathcal{R}(\mathcal{H}_2) \leq \kappa_2^2 \tilde{A}_2 + 2\kappa_2(\tilde{A}_2 \mathcal{R}(\mathcal{H}_2))^{1/2}.$$

*Remark* 6. If we consider deep models and keep the $\mathcal{H}_1, \mathcal{H}_2$ the same but iterate over layers $j$ stitching forward, then

$$\mathcal{R}(\mathcal{H}_1) - \mathcal{R}(\mathcal{H}_2) \leq \min_j \left\{ (\kappa_2^{(j)})^2 \tilde{A}_2^{(j)} + 2\kappa_2^{(j)} (\tilde{A}_2^{(j)} \mathcal{R}(\mathcal{H}_2))^{1/2} \right\}.$$

Alternatively, by making similar assumptions and swapping the index $1 \leftrightarrow 2$, which requires $\mathcal{G}_1 = \mathcal{G}_2$ up to a linear layer (due to the $\mathcal{S}_{1,2} \circ \mathcal{G}_2 \subseteq \mathcal{G}_1$ condition), we get

$$|\mathcal{R}(\mathcal{H}_1) - \mathcal{R}(\mathcal{H}_2)| \leq \max_{i \in \{1,2\}} \left\{ \kappa_i^2 \tilde{A}_i + 2\kappa_i(\tilde{A}_i \mathcal{R}(\mathcal{H}_q))^{1/2} \right\}.$$

The above result can be stated informally as "alignment at similar depth (measured backward from the output) bounds differences in risk".

The results presented have several practical implications. First, we build on the experiments from Huh et al. (2024), which provide evidence for the alignment of deep networks at a large scale using measures similar to kernel alignment. By establishing a connection between kernel alignment and stitching, our work supports building universal models that share architectures across modalities as scale increases. Second, we can elucidate the experiments from Bansal et al. (2021), which suggest that typical SGD minima have low stitching costs (stitching connectivity). This aligns with works that argue feature learning under SGD can be understood through the lens of adaptive kernels (Radhakrishnan et al., 2022; Atanasov et al., 2022).

## 5 CONCLUSION

In this paper, we review and unify several representation alignment metrics, including kernel alignment, distance alignment, and independence testing, demonstrating their equivalence and interrelationships. Additionally, we formalize the concept of stitching, a technique used in uni/multi-modal settings to quantify alignment in relation to a given task. Furthermore, we establish bounds on stitching error across different modalities and derive stitching error bounds based on misalignment, along with their generalizations and implications.

### ACKNOWLEDGMENT

L.R. is thankful to the CBMM-Hebrew University workshop organizers and Philipp Isola for the talk that inspired this work. L. R. acknowledges the financial support of the European Research Council (grant SLING 819789), the European Commission (Horizon Europe grant ELIAS 101120237), the US Air Force Office of Scientific Research (FA8655-22-1-7034), the Ministry of Education, University and Research (FARE grant ML4IP R205T7J2KP; grant BAC FAIR PE00000013 funded by the EU - NGEU). This work represents only the view of the authors. The European Commission and the other organizations are not responsible for any use that may be made of the information it contains.

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

## 6 APPENDIX

### 6.1 ALIGNMENT TO TASK

Here we mention ideas of alignment between a representation and task used to estimate generalization error and characterize spectral contributions to sample complexity.

**Kernel alignment risk estimator (KARE)** In Jacot et al. (2020) we have the following definition for KARE which is an estimator for risk.

$$\rho(\lambda, y_n, K_n) = \frac{\frac{1}{n}\langle (K_n/n + \lambda I)^{-2}, y_n y_n^T\rangle}{(\frac{1}{n}\text{Tr}[(K_n/n + \lambda I)^{-1}])^2}$$

This was also obtained in Golub et al. (1979), Wei et al. (2022), Craven & Wahba (1978).

**Spectral task-model alignment** From Canatar et al. (2021), we have a definition for the cumulative power distribution which quantifies task-model alignment.

$$C(n) = \frac{\sum_{i \leq n} \eta_i w_i^2}{\sum_i \eta_i w_i^2}$$

Here $K = \sum_i \eta_i \phi_i \otimes \phi_i$, $\langle \phi_i, \phi_j \rangle = \delta_{i,j}$, and target $h_\mu = \sum_i w_i \sqrt{\eta_i}\phi_i$. $C(n)$ can be interpreted as fraction of variance of $h_\mu$ explained by first $n$ features. The faster $C(n)$ goes to 1, the higher the alignment.

**Source Condition** From Rosasco et al. (2005) we have bounds on generalization of kernel ridge assuming some regularity of $h_\mu$, called source condition

$$h_\mu \in \Omega_{r,R} = \left\{ h \in L^2(X, \rho) : h = L_K^r v, \|v\|_K \leq R \right\}$$

Assuming $h_\mu = \sum_i w_i \sqrt{\eta_i}\phi_i$, then the statement can be rewritten as

$$\sum_{i=1}^{\infty} \frac{\eta_i w_i^2}{\eta_i^{2r}} < \infty$$

*Remark* 7. KTA appears in several theoretical applications. Cristianini et al. (2001) bounds generalization error of Parzen window classifier [1]. Cristianini et al. (2006); Cortes et al. (2012) show that there exist predictors for which kernel target alignment (KTA) $A(K, yy^T)$ bounds risk.

$$h(x) = \frac{\mathbb{E}_{x',y'}\left[K(x,x')y'\right]}{\mathbb{E}_{x',x}\left[K(x,x')^2\right]} \Rightarrow \mathcal{R}(h) \leq 2(1 - A(K, yy^T))$$

Furthermore, several authors including Atanasov et al. (2022); Paccolat et al. (2021); Kopitkov & Indelman (2020); Fort et al. (2020); Shan & Bordelon (2021) use KTA to study feature learning and Neural Tangent Kernel evolution.

---

[1]Cortes et al. (2012) notes error in proof since implicitly assumes $\max_x \mathbb{E}_{x'}\left[K^2(x,x')\right]/\mathbb{E}_{x,x'}\left[K^2(x,x')\right] = 1$ making kernel constant. However proof can be saved with an additional assumption.

## 6.2 OTHER NOTIONS FOR ALIGNMENT FROM INDEPENDENCE TESTING

**Constrained Covariance (COCO)** Then Gretton et al. (2005a) proposed the concept of constrained covariance as the largest singular value of the cross-covariance operator,

$$\text{COCO}(\mu, \mathcal{H}_1, \mathcal{H}_2) = \sup\{\text{cov}[h_1(x_1), h_2(x_2)] : h_1 \in \mathcal{H}_1, h_2 \in \mathcal{H}_2\}$$

**Kernel Canonical Correlation (KCC)** From Bach & Jordan (2002)

$$\text{KCC}(\mu, \mathcal{H}_1, \mathcal{H}_2, \kappa) = \sup\left\{\frac{\text{cov}[h_1(x_1), h_2(x_2)]}{(\text{var}(h_1(x_1)) + \kappa\|h_1\|_{\mathcal{H}_1}^2)^{1/2}(\text{var}(h_2(x_2)) + \kappa\|h_2\|_{\mathcal{H}_2}^2)^{1/2}} : h_1 \in \mathcal{H}_1, h_2 \in \mathcal{H}_2\right\}$$

The next two are bounds on mutual information from correlation and covariance respectively

**Kernel Mutual Information (KMI)** From Bach & Jordan (2002)

$$\text{KMI}(\mathcal{H}_1, \mathcal{H}_2) = -\frac{1}{2}\log(|I - (\kappa_{1,n}\kappa_{2,n})K_{1,n}K_{2,n}|)$$

where kernels are centered and $\kappa_{q,n} = \min_i \sum_j K_q(x_{q,i}, x_{q,j})$ but empirically $\kappa = 1/n$ suffices.

## 6.3 ADDITIONAL PROOFS

In this section, we provide the detailed proofs of Lemmas presented in and Section 3 and Section 4. We begin with the proof of Lemma 1. For completeness, we first restate the lemma below.

**Lemma 4.** *Assume* $|K_q(x_q, x_q')| \leq C_q$. *Let* $\hat{A}_{1,2}(X) = \hat{A}_{1,2}((x_1^1, x_2^1), \ldots, (x_1^n, x_2^n)) = \frac{1}{n^2}\langle K_1, K_2\rangle_F$. *Let* $A_{1,2} = \mathbb{E}\hat{A}_{1,2}$, $\hat{A} = \frac{\hat{A}_{1,2}}{\sqrt{\hat{A}_{1,1}\hat{A}_{2,2}}}$, *and* $A = \frac{A_{1,2}}{\sqrt{A_{1,1}A_{2,2}}}$. *Then with probability at least* $1 - \delta$, *and* $\epsilon = \sqrt{(32/n)\log(2/\delta)}$, *we have* $|\hat{A} - A| \leq C(X)\epsilon$, *where* $C(X)$ *is non-trivial function.*

*Proof.* Let $(x_1^{i'}, x_2^{i'}) = (x_1^i, x_2^i)$ for all $i = 1, \ldots n$ except $k$. Then

$D_{ij} = K_1(x_1^i, x_1^j)K_2(x_2^i, x_2^j) - K_1(x_1^{i'}, x_1^{j'})K_2(x_2^{i'}, x_2^{j'})$ and note $|D_{ij}| \leq 4C_1C_2$. Then

$$|\hat{A}_{1,2}(X) - \hat{A}_{1,2}(X')| = n^{-2}\left(2\sum_{j \neq i}|D_{ij}| + |D_{ii}|\right) \leq 4C_1C_2\frac{2n-1}{n^2} \leq \frac{8C}{n}$$

Applying McDiarmid, we get

$$P\{|\hat{A}_{1,2} - A_{1,2}| \geq \epsilon\} \leq 2\exp\left(\frac{-\epsilon^2 n}{32C^2}\right)$$

which can also be read as, with probability at least $1 - \delta$, $|\hat{A}_{1,2} - A_{1,2}| \leq \epsilon = \sqrt{(32/n)\log(2/\delta)}$

Finally, we show that $|\hat{A}_{i,j} - A_{i,j}| \leq \epsilon$ for $i, j \in \{1, 2\}$ gives $|\hat{A} - A| \leq C(X)\epsilon$.

$$|\hat{A} - A| = \left|\hat{A}_{1,2}(\hat{A}_{1,1}\hat{A}_{2,2})^{-1/2} - A_{1,2}(A_{1,1}A_{2,2})^{-1/2}\right|$$

$$= |\hat{A}_{1,2} - A_{1,2}|(\hat{A}_{1,1}\hat{A}_{2,2})^{-1/2} + A_{1,2}\left|(\hat{A}_{1,1}\hat{A}_{2,2})^{-1/2} - (A_{1,1}A_{2,2})^{-1/2}\right|$$

$$= |\hat{A}_{1,2} - A_{1,2}|(\hat{A}_{1,1}\hat{A}_{2,2})^{-1/2}$$

$$+ A_{1,2}\left(\left|(\hat{A}_{1,1}\hat{A}_{2,2})^{-1/2} - (A_{1,1}\hat{A}_{2,2})^{-1/2}\right| + \left|(A_{1,1}\hat{A}_{2,2})^{-1/2} - (A_{1,1}A_{2,2})^{-1/2}\right|\right)$$

Lastly, we can use

$$(x^{-1/2} - y^{-1/2}) = \frac{y^{1/2} - x^{1/2}}{(xy)^{1/2}} = \frac{y - x}{(xy)^{1/2}(y^{-1/2} + x^{-1/2})}$$

□

Now we are in the position to prove Lemma 2. For completeness, we first restate the lemma below.

**Lemma 5.** *Suppose $dim(\mathcal{Y}_1) = dim(\mathcal{Y}_2) = d$ and $\mathcal{R}_1 = \mathcal{R}_2$. Let $g_q \in \mathcal{G}_q$ be linear with $g_q(z_q) = W_q z_q$ and $W_q \in \mathbb{R}^{d \times d_q}$. Let $s_{1,2} : \mathcal{Z}_1 \to \mathcal{Z}_2$ be linear with $s_{1,2}(z_1) = S_{1,2} z_1$ and $S_{1,2} \in \mathbb{R}^{d_2 \times d_1}$. Then $\mathcal{R}_{1,2}^{\text{stitch}}(\mathcal{S}_{1,2}) = \mathcal{R}_1(\mathcal{H}_1)$.*

*Proof.* For the linear case, there exists a vector $W_q \in \mathbb{R}^{d \times d_q}$, such that $g_q(z_q) = W_q z_q$, $z_q \in \mathbb{R}^{d_q}$. We can write the error of stitching as

$$\begin{aligned}
\mathcal{R}_{1,2}^{\text{stitch}}(s_{1,2}) &= \mathbb{E}\left[\|W_2 S_{1,2} f_1 - y\|^2\right] \\
&= \mathbb{E}\left[\|(W_2 S_{1,2} - W_1) f_1\|^2\right] + \mathbb{E}\left[\|W_1 f_1(x) - y\|^2\right] \\
&= \|W_2 S_{1,2} - W_1\|_{\eta_1}^2 + \mathcal{R}_1(h_1),
\end{aligned}$$

where we used that for $W_1$ to be optimal, we require $\partial_{W_1} \mathcal{R}_1(h_1) = \mathbb{E}\left[(W_1 f_1 - y) f_1^T\right] = 0$. Minimizing with respect to $S_{1,2}$ yields

$$\mathcal{R}_{1,2}^{\text{stitch}}(\mathcal{S}_{1,2}) = \|\Pi_2^{\perp} W_1\|_{\eta_1}^2 + \mathcal{R}_1(\mathcal{H}_1),$$

where we use $\Pi_2 = I - (W_2^T \text{diag}(\eta_1) W_2)^{\dagger} W_2^T \text{diag}(\eta_1)$ to denote the residual of the generalized $\eta_1$-projection onto (column) span of $W_2$. We note that in general, as long as $d \le d_2$, we have $\mathcal{R}_{1,2}^{\text{stitch}}(\mathcal{S}_{1,2}) = \mathcal{R}_1(\mathcal{H}_1)$. □

