# OpenReview forum: "Towards a learning theory of representation alignment"
_ICLR.cc/2025/Conference — ICLR 2025 Poster_

### Official Review · Reviewer_EzC8 · 2024-10-31

**Soundness:** 2
**Presentation:** 3
**Contribution:** 2
**Rating:** 5
**Confidence:** 2

**Summary:**

This work proposes a theoretical foundation for understanding representations alignment, especially the alignments for multimodal models.
The main contribution is in the idea of connecting different representations through a so-called "stitching" mechanism.

**Strengths:**

The paper is generally well-written. The authors gave an excellent review of measures of representation alignment, which is a nice entry point for understanding the paper.
This paper is also supported by a thorough mathematical argument about the stitching error between models.

**Weaknesses:**

1. The paper provides no empirical experiment demonstrating the proposed theory's correctness.
2. Only one type of stitching, Linear Stitching, is argued. It would be helpful if the authors could briefly explain other stitching types, as this will help to familiarize the readers to this idea.

**Questions:**

1. The reviewer understands that the authors try to avoid complexity by only arguing about linear stitching. But how is the generality of the linear stitching? And what is its limitation? Do different types of representations require different stitching?  Please add this argument in the discussion part of the paper.

2. Please provide at least one experiment to demonstrate the correctness of the proposed theory, for example, through transfer learning.

---

> ### Author Response · Authors · 2024-11-27
>
> We thank the reviewer for the thoughtful feedback and reply to the comments below.
>
> > Only one type of stitching, Linear Stitching, is argued. It would be helpful if the authors could briefly explain other stitching types, as this will help to familiarize the readers to this idea.The reviewer understands that the authors try to avoid complexity by only arguing about linear stitching. But how is the generality of the linear stitching? And what is its limitation? Do different types of representations require different stitching? Please add this argument in the discussion part of the paper.
>
> We now include a discussion of the linearity assumptions and more motivation for simple stitching maps in general, as follows:
>
> - The functions in $S_{1,2}$ are typically simple maps such as linear layers or convolutions of size one, to avoid introducing any learning, as emphasized in Bansal et al. (2021). The aim is to measure the compatibility of two given representations without fitting a representation to another. One perspective inspired by Lenc & Vedaldi (2015) is that we should not penalize certain symmetries, such as rotations, scaling, or translations, which do not alter the information content of the representations. Furthermore, the amount of unwanted learning may be quantified by stitching from a randomly initialized network.
>
> - In arguing that kernel alignment bounds stitching error for Theorem 1, we made several simplifying assumptions, which we now assess. Firstly, we restricted the stitching $S_{1,2}$ to linear maps, based on the transformations used in practice (Bansal et al., 2021; Csisz´ arik et al., 2021), and to preserve the significance of the original representations. If we relax the assumption, we note that we would get a similar result, with $\tilde A_2 = \inf_{s_{1,2} \in \mathcal S_{1,2}} E[\| s_{1,2}(f_1(x))−f_2(x)\|^2]$. Interestingly, for $s_{1,2}$ to use only information about the covariance of $f_1,f_2$, similarly to kernel alignment, $s_{1,2}$ must be linear. Furthermore, we note that for stitching classes that include all linear maps, the linear result holds.
>
> >The paper provides no empirical experiment demonstrating the proposed theory's correctness. Please provide at least one experiment to demonstrate the correctness of the proposed theory, for example, through transfer learning.
>
>  The primary objective of this paper is to establish a theoretical framework for studying and comparing learned representations across models and tasks. We agree that exploratory experiments could yield insights into when kernel alignment and stitching error correlate or diverge and we will add the experiments in the final revision.
>
> While stitching and transfer learning both involve blending two models, the two differ since stitching only learns the stitching layer while transfer learning requires more learning (for example fine-tuning typically involves training at least the second model). Yet, for a simple connection, we can consider measuring kernel alignment between the representation of model 1 and the task of model 2. Then the misalignment works as a upper bound for the risk of the fine-tuned model. Investigating stronger relationship between stitching, kernel alignment, and transfer learning is a interesting potential direction for future exploration.
>
> **References:**
>
> [1] Yamini Bansal, Preetum Nakkiran, and Boaz Barak. Revisiting model stitching to compare
> neural representations. Advances in Neural Information Processing Systems, 34:225–236, 2021.
>
> [2] Karel Lenc and Andrea Vedaldi. Understanding image representations by measuring their equivariance and equivalence. In Proceedings of the IEEE Conference on Computer Vision and Pattern
> Recognition, pp. 991–999, 2015.

---

> > ### Comment · Reviewer_EzC8 · 2024-12-02
> > **response to official comment by authors**
> >
> > Thank you for adding some discussions to the revised paper.
> > Unfortunately, the authors did not provide any experiments in the final revision of their paper.

---

> > > ### Author Response · Authors · 2024-12-04
> > >
> > > We thank the reviewer for the additional feedback and comment further on an experiment and on transfer learning.
> > >
> > > Consider the following setup. For $q=1,2$, let $h_q(x) = \tanh\left( \sqrt{\frac{2}{k_q}} f_q(x)\cdot a_q\right)$, where $f_{q,i}(x) = \cos(w_{q,i}^T x/\lambda_q + \phi_{q,i})$, $k_q$ is the number of random features, and $w_{q,i}\sim \mathcal N(0,1)$, $\phi_{q,i}\sim U[0,2\pi]$. $f_{q,i}$ represent random Fourier features which approximate a Gaussian kernel with bandwidth $\lambda_q$, and $g_q$ is the composition of linear transformation and $\tanh$ function. Additionally, we can assume gaussian input data $x\sim \mathcal N(0, I_d)$.
> > >
> > > Suppose we fix the second model, which we'll consider as the target ($y=h_2(x)$), and vary the first by selecting different bandwidths $\lambda_1$. We can then compute and compare different measures of alignment including empirical alignment between $K_q = \frac{2}{k_q}f_q f_q^T$, excess stitching risk $\min_{S\in\mathbb R^{k_2\times k_1}}\hat {\mathbb{E}}[(y - g_2(S f_1(x))^2]$, as well as the modified notion of alignment emerging from our results $\min_{S\in\mathbb R^{k_2\times k_1}}\hat {\mathbb{E}}[||f_2 - S f_1(x)||^2]$.
> > >
> > > The experiment has been run for $d=1$ and shows correlation between kernel misalignment and stitching error. We will conduct further analyses, varying the sources of misalignment and considering datasets like MNIST, and will report results.
> > >
> > > Finally, regarding transfer learning, we note that when stitching to the output layer, the excess stitching risk is the same as "linear transferability" (up to sign).

---

### Official Review · Reviewer_YJw7 · 2024-11-03

**Soundness:** 2
**Presentation:** 3
**Contribution:** 3
**Rating:** 6
**Confidence:** 3

**Summary:**

The paper proposes a theoretical perspective on representation alignment of AI models.
The authors state  that while there is empirical work  that supports the idea of representation alignment for bigger models, there is little theoretical perspective on the idea of representational alignment. The authors focus on the practice of (model) stitching as a device to study representational alignment. More specifically,
they first present different ways to measure representation alignment and then focus on stitching as task to contextualise a theory for representational alignment.
The contribution of this paper is showing that a stitching error and stitching error bound can be derived based on the kernel alignment of the underlying representations.

Overall, I think this is a good paper attempting to provide a theoretical framework to representational alignment which is usually an empirical field of research. The derived stitching error and stitching error bound seem important in order to quantify generalisation error of different models. To improve the submission I would suggest making Section 4 more coherent with Section 3, since at the moment they seem a bit disconnected to me notation wise. I would also suggest either adding a small experiment showing the usefulness of the derived theory or contextualising it within current empirical methods.

**Strengths:**

- The authors do a good job of introducing the different concepts of kernel alignment and independence testing needed for their theory.
- Using Kernel Alignment to provide a generalisation error bound for stitching seems novel and useful given the rise of representation learning.
- I think the paper is generally well written making it accessible for readers that are not that familiar with a learning theoretic perspective on this problem.

**Weaknesses:**

- It is unclear to me from Section 4 how the stitching error and stitching error bound can be used. Since the paper makes reference to existing empirical work, I would like to see a more concrete example or a reference to where the derived theory fits in existing empirical work.
- For instance how does the proposed theory quantify the stitching error in current approaches (e.g., can different layers be compared equivalently, does normalisation, regularisation play a role?).
- I feel there is a slight disconnect between sections 3 and 4. While in section 3 a comprehensive listing of ways to measure representation alignment is presented, the introduced notation is not really used in Section 4. This makes it a bit difficult to follow what parts of Section 3 inspired Section 4. Perhaps you could write a small paragraph at the beginning of Section 4 stating how Section 3 serves as building blocks for deriving the stitching error and stitching error bound in Section 4.

**Questions:**

- Would it be possible to design a small experiment with, e.g., MNIST where the stitching error (bound) is used to quantify the possible generalisation error?
- Alternatively you could add a small discussion on how your theory fits in with current stitching methods or how it could be applied to learn more better representations.

---

> ### Author Response · Authors · 2024-11-27
>
> We thank the reviewer for the thoughtful feedback and reply below.
>
> >It is unclear to me from Section 4 how the stitching error and stitching error bound can be used. Since the paper makes reference to existing empirical work, I would like to see a more concrete example or a reference to where the derived theory fits in existing empirical work.
>
> We now include a discussion of practical implications of the stitching bound at the end of Section 4, connecting to the introduction and empirical works in the revised manuscript:
>
> - First, we can build on the experiments from Huh et al. (2024) which show evidence for the alignment of deep networks at large scale using alignment measures similar to kernel alignment. By connecting kernel alignment to stitching, our work supports building universal models sharing architecture across modalities as scale increases  as stitching error could be bounded by the misalignment.
>
> - Second, we provide support to the experiments from Bansal et al. (2021) which suggest that typical SGD minima have low stitching costs (stitching connectivity) through works that argue feature learning under SGD can be understood via adaptive kernels (Radhakrishnan et al., 2022; Atanasov et al., 2021).
>
> > For instance how does the proposed theory quantify the stitching error in current approaches (e.g., can different layers be compared equivalently, does normalisation, regularisation play a role?).
>
> In our work, we use the stitching error to measure the alignment of learned representations at two given layers in two models via spectral decompositions of the kernel alignment. Our primary objective is not to learn representations but to assess how well-aligned the existing representations are.
>
> That said, exploring how different layers or approaches (normalization, regularization, etc.) influence feature alignment and affect the stitching error is a fascinating direction of research, though it is not our main focus here. Lenc & Vedaldi (2015) demonstrated that the early layers of convolutional networks tend to be more compatible compared to later layers, which are increasingly task-specific. Investigating the stitching error across various “stitching points” could offer valuable insights to get better feature spaces and presents an interesting direction for future studies.
>
> >I feel there is a slight disconnect between sections 3 and 4. While in section 3 a comprehensive listing of ways to measure representation alignment is presented, the introduced notation is not really used in Section 4. This makes it a bit difficult to follow what parts of Section 3 inspired Section 4. Perhaps you could write a small paragraph at the beginning of Section 4 stating how Section 3 serves as building blocks for deriving the stitching error and stitching error bound in Section 4.
>
> We reorganized the end of Section 3 and start of Section 4 to make the connection more explicit.
>
> > Would it be possible to design a small experiment with, e.g., MNIST where the stitching error (bound) is used to quantify the possible generalisation error?
>
> The primary objective of this paper is to establish a theoretical framework for studying and comparing learned representations across models and tasks.   We agree that exploratory experiments could yield insights into when kernel alignment and stitching error correlate or diverge and we will add the experiments in the final revision.
>
> > Alternatively you could add a small discussion on how your theory fits in with current stitching methods or how it could be applied to learn more better representations.
>
> For fitting in with empirical works, please refer to the response provided for the first question above.
>
> For better representations learning:
> We want to emphasize that stitching is not to learn further representations. But future exploratory experiments could yield insights into when kernel alignment and stitching error correlate or diverge. A possible extension of this idea could involve designing training objectives that explicitly promote feature alignment or incorporating stitching-based evaluation metrics to iteratively refine representations.
>
> **References:**
>
> [1] Atanasov et al., (2021): Neural networks as kernel learners: The silent alignment effect.
>
> [2] Bansal et al. (2021): Revisiting model stitching to compare neural representations.
>
> [3] Huh et al. (2024): Position: The platonic representation hypothesis.
>
> [4] Lenc & Vedaldi (2015): Understanding image representations by measuring their equivariance and equivalence.
>
> [5] Radhakrishnan et al., (2022): Mechanism of feature learning in deep fully connected networks and kernel machines that recursively learn features.

---

> > ### Comment · Reviewer_YJw7 · 2024-12-02
> >
> > I appreciate the authors' clarifications and improvements to the written sections.
> >
> > Unfortunately, as the other reviewers also point out, I am still missing a concise experiment or at least more insight on how the proposed theory is useful or how it could be implemented to study current algorithms. While the authors added a paragraph in the paper regarding this issue, it reads more like related work than an actual analysis into existing methods.
> >
> > I therefore for now will be keeping my score.

---

### Official Review · Reviewer_smSY · 2024-11-03

**Soundness:** 3
**Presentation:** 3
**Contribution:** 3
**Rating:** 6
**Confidence:** 2

**Summary:**

The authors compiled broad notions of representational alignment metrics and provide connected interpretation. Next, they mathematically formulate stitching method, which is used to evaluate similarity of representations between different models given a task by plugging representation from one model to another. They provide the generalization error bound of linearly stitched model based on kernel alignment.

**Strengths:**

While I am not familiar with all the references,the first section of the paper where the authors make a overview of different formulation of representation alignment and provide connected interpretation from different community seems to be a nice contribution.

The theoretical formulation of stitching method that is used frequently in practice is relevant point and the results can give valuable insights to use of stitching method in practice

In general the paper is well-written and well structured.

**Weaknesses:**

- I appreciate the first section of the paper, but it would be nice to have the last paragraph where authors could summarize and make a brief overview of everything in one place.

- While it being a solid theoretical work, I think it is great that the settings in questions are still relevant for practice. Having a couple of sentences summarizing practical implication of the results in conclusion might be helpful for broader audience.

- It is little confusing about significance of 'task defined representation' and 'modality specific representation' in stitching setting.

Minor comments on the manuscript
- little confused by term 'representation $\mathcal{H}_q$' online 4 as I thought it's entire input-output mapping
- I think there are some indexing, sqrt typos on "Spectral interpreation of KA" section; what is $\rho$?
- Typo on line 494

**Questions:**

- In theorem 3, why does swapping the index 1,2 require $\mathcal{G}_1 = \mathcal{G}_2$ up to linear?

- On remark 3, could you elaborate little more on the regularization of $S_{1,2}$?

- It looks like the stitching setting is quite relevant to transfer learning. Could the current result tell anything about kernel alighment and its relevance on transfer learning perspective?

---

> ### Author Response · Authors · 2024-11-27
>
> We thank the reviewer for the careful reading and thoughtful feedback and reply to the comments below.
>
> > I appreciate the first section of the paper, but it would be nice to have the last paragraph where authors could summarize and make a brief overview of everything in one place.
>
> We now include a sentence to highlight the purpose of the section and provide and overview, as follows:
> - In summary, we've introduced several popular measures for alignment between two representations and related them via spectral decompositions to a central notion of kernel alignment generalized for RKHS.
> Similar notions can be used to measure alignment between a model and a task to estimate generalization error.
>
> > While it being a solid theoretical work, I think it is great that the settings in questions are still relevant for practice. Having a couple of sentences summarizing practical implication of the results in conclusion might be helpful for broader audience.
>
> We now include a discussion of practical implications of the stitching bound at the end of Section 4, connecting to the introduction and empirical works in the revised manuscript:
>
> - First, we can build on the experiments from Huh et al. (2024) which show evidence for the alignment of deep networks at large scale using alignment measures similar to kernel alignment. By connecting kernel alignment to stitching, our work supports building universal models sharing architecture across modalities as scale increases  as stitching error could be bounded by the misalignment.
>
> - Second, we provide support to the experiments from Bansal et al. (2021) which suggest that typical SGD minima have low stitching costs (stitching connectivity) through works that argue feature learning under SGD can be understood via adaptive kernels (Radhakrishnan et al., 2022; Atanasov et al., 2021).
>
> > It is little confusing about significance of 'task defined representation' and 'modality specific representation' in stitching setting.
>
> Stitching involves finding a transformation to align two representations. These representations are often task-dependent, meaning they are shaped by the training process for specific tasks (classifying object from different modalities). When stitching aligns these representations, it is implicitly evaluating how compatible the task-relevant features are. When we say ``task aware representation," it means the outputs (task) $y$ are incorporated into measuring the alignment and it usually relates to generalization error. In particular, it is interesting to consider the similarity between two representations for different modalities trained for the same task.
>
> We also reorganized the end of Section 3 and the start of Section 4 so the significance should be more explicit.
>
> > little confused by term 'representation $\mathcal{H}_q$' online 4 as I thought it's entire input-output mapping
>
> Fixed. It should be ``model $\mathcal{H}_q$" instead of ''representation $\mathcal{H}_q$"
>
> > I think there are some indexing, sqrt typos on "Spectral interpreation of KA" section; what is $\rho$?
>
> Fixed. There were typos in the indices, but not in powers/sqrt.
>
> > Typo on line 494
>
> Fixed
>
> > In theorem 3, why does swapping the index 1,2 require $\mathcal G_1 = \mathcal G_2$ up to linear?
>
> This is due to the $\mathcal S_{1,2} \circ \mathcal G_2 \subseteq \mathcal G_1$ condition.
>
> > On remark 3, could you elaborate little more on the regularization of $\mathcal S_{1,2}$?
>
> Regularization as mentioned in remark 3 is analogous to regularization in linear regression, which is now made more explicit in the manuscript.
>
> > It looks like the stitching setting is quite relevant to transfer learning. Could the current result tell anything about kernel alignment and its relevance on transfer learning perspective?
>
> While stitching and transfer learning both involve blending two models, the two differ since stitching only learns the stitching layer while transfer learning requires more learning (for example fine-tuning typically involves training at least the second model). Yet, for a simple connection, we can consider measuring kernel alignment between the representation of model 1 and the task of model 2. Then the misalignment works as an upper bound for the risk of the fine-tuned model. Investigating stronger relationship between stitching, kernel alignment, and transfer learning is an interesting potential direction for future exploration.
>
> **References:**
>
> [1] Atanasov et al., (2021): Neural networks as kernel learners:
> The silent alignment effect.
>
> [2] Bansal et al. (2021): Revisiting model stitching to compare
> neural representations.
>
> [3] Huh et al. (2024): Position: The platonic representation hypothesis.
>
> [4] Radhakrishnan et al., (2022): Mechanism of feature learning in deep fully connected networks and kernel machines that recursively learn features.

---

> ### Comment · Reviewer_smSY · 2024-12-02
>
> Thank you for the clarification, I am convinced that the paper tackles interesting question providing a theoretical framework to understand the topic. Given the response of the authors, I think there are literature which provides empirical findings in line with the theory, although having more direct exhibition of the theory via additional experiment might strengthen the paper.
>
> I will keep my score, but this is mostly because I cannot award 7.

---

### Official Review · Reviewer_embS · 2024-11-05

**Soundness:** 4
**Presentation:** 3
**Contribution:** 3
**Rating:** 6
**Confidence:** 2

**Summary:**

In this manuscript the authors first give a review of various measures of representation alignment, with a particular focus on showing how they relate. They show that many different approaches can be straightforwardly related to kernel alignment.

The authors then formalise the stitching approach to measuring task-relevant representation alignment. They demonstrate that bounds on stitching error can be derived and related to the alignment metrics that they reviewed in the first section.

**Strengths:**

The manuscript is clear and well-written.

Although perhaps a little dense, the first section does a good job of introducing various notions of representation alignment and elucidating their relationships. I found this precise and succinct presentation to be useful and digestible.

The formalisation of stitching methods is also clear and straightforward. While the assumption of linear stichers is quite restrictive, it does lead to some nice proofs.

**Weaknesses:**

I don’t think the manuscript has any significant weaknesses.

I do think that at the end of section 3 it might be helpful to give a few sentence summary of the relationships between the various measures of alignment. While the preceding material is complete, it would help readers who perhaps have not digested all of that material on a first pass to more easily comprehend the manuscript.

As mentioned above, the restriction to linear stitching functions is potentially quite limiting. It might be nice to comment more on the impact of this assumption on the analysis.

It should be noted that I am not very familiar with the field of representation alignment, so I am not in a position to comment on this work relative to the existing literature. I will have to leave that to the other reviewers. My score reflects this uncertainty, but it should be understood that this is not a comment on the manuscript, but rather my uncertainty about its impact, novelty, and relevance.

**Questions:**

Just a few nits:

Figure 1: it would nice to note for the reader that these symbols will be defined in section 2.

L130: I’m not familiar with the notation using the #. Perhaps add a note or reference, or a an explanatory word or two in the sentence where it is first used?

L188: I didn’t understand why H_q had codomain \mathbb{R}. This seems to be assuming something about Y_q that up until this point hasn’t been assumed. Could you clarify?

L241 & L670: McDiarmin -> McDiarmid!

L351: modals -> models?

---

> ### Author Response · Authors · 2024-11-27
>
> We thank the reviewer for the thoughtful feedback and reply to the comments below.
>
> >  I do think that at the end of section 3 it might be helpful to give a few sentence summary of the relationships between the various measures of alignment. While the preceding material is complete, it would help readers who perhaps have not digested all of that material on a first pass to more easily comprehend the manuscript.
>
> We include a sentence to highlight the purpose of this section at the end of section 3 in the revised draft:
> - In summary, we've introduced several popular measures for alignment between two representations and related them via spectral decompositions to a central notion of kernel alignment generalized for RKHS.
> Similar notions can be used to measure alignment between a model and a task to estimate generalization error.
>
> > As mentioned above, the restriction to linear stitching functions is potentially quite limiting. It
> might be nice to comment more on the impact of this assumption on the analysis.
>
> We include a discussion on the linearity assumption and more motivation for simple stitching maps in general in the revised manuscript, as follows:
>
> - The functions in $S_{1,2}$ are typically simple maps such as linear layers or convolutions of size one, to avoid introducing any further representation learning, as emphasized in Bansal et al. (2021). The goal is to measure the compatibility of two given representations without "fitting" a representation to the other. Another perspective,  inspired by Lenc & Vedaldi (2015), is that we should not penalize certain symmetries, such as rotations, scaling, or translations, which do not alter the information content of the representations. Further, the amount of unwanted learning may be quantified by stitching from a randomly initialized network.
>
> - In arguing that kernel alignment provide a bound on the stitching error  (Theorem 1), we made several simplifying assumptions, which we assess next. Firstly, we restricted the stitching $S_{1,2}$ to linear maps, based on the transformations used in practice (Bansal et al., 2021; Csisz´ arik et al., 2021), and to preserve the significance of the original representations. If we relax this assumption, we note that we would get a similar result, with $\tilde A_2 = \inf_{s_{1,2} \in \mathcal S_{1,2}} E[\| s_{1,2}(f_1(x))−f_2(x)\|^2]$. Interestingly, for $s_{1,2}$ to use only information about the covariance of $f_1,f_2$, similarly to kernel alignment, $s_{1,2}$ must be linear. Further, we note that for stitching approaches that include all linear maps, the linear result holds.
>
>
>
>
> >Q1 Figure 1: it would nice to note for the reader that these symbols will be defined in section 2.
>
> We've added one sentence to improve the readability in the new draft.
>
> > Q2 L130: I’m not familiar with the notation using the #. Perhaps add a note or reference, or a an explanatory word or two in the sentence where it is first used?
>
> We now include a footnote describing the "pushforward" map # on page 3.
>
> > Q3  L188: I didn’t understand why H_q had codomain \mathbb{R}. This seems to be assuming something about Y_q that up until this point hasn’t been assumed. Could you clarify?
>
> We clarify this point in the revision as follows:
> -  After introducing the definition for Empirical Kernel Alignment, in this paragraph we wanted to draw a connection to RKHS theory. To get a consistent definition, it suffices to consider outputs in one dimension. We can then generalize the definition to higher dimensions considering each output component separately (separable matrix-valued RKHS).
>
>
> >Q4 L241 & L670: McDiarmin -> McDiarmid!
>
> > Q5 L351: modals -> models?
>
> We've fixed these typos in the draft. Thanks for careful reading!
>
>
> **References:**
>
> [1] Yamini Bansal, Preetum Nakkiran, and Boaz Barak. Revisiting model stitching to compare neural
> representations. Advances in Neural Information Processing Systems, 34:225–236, 2021.
>
> [2]  Karel Lenc and Andrea Vedaldi. Understanding image representations by measuring their equivariance and equivalence. In Proceedings of the IEEE Conference on Computer Vision and Pattern Recognition, pp. 991–999, 2015.
>
> [3] Adri´an Csisz´arik, P´eter K˝or¨osi-Szab´o, Akos Matszangosz, Gergely Papp, and D´aniel Varga. Similarity and matching of neural network representations. Advances in Neural Information Processing
> Systems, 34:5656–5668, 2021.

---

### Meta-Review · Area_Chair_nbrh · 2024-12-24

**Metareview:**

The paper reviews existing representation alignment techniques and provides a theoretical analysis of representation "stitching" for different modalities.  The strengths of the paper are the nice survey of representation alignment techniques and novel theory regarding stitching.  The main weakness is the absence of any experiments.  Nevertheless, the contributed theory is interesting and worthwhile.  Hence this represents a valuable contribution to the understanding of representation alignment.

**Additional Comments On Reviewer Discussion:**

The reviewers discussed whether the paper should include experiments.  While there wasn't a consensus, ICLR does welcome theory papers.

---

### Decision · Program_Chairs · 2025-01-22

Accept (Poster)